# Characterization of Novel *Trypanosoma cruzi*-Specific Antigen with Potential Use in the Diagnosis of Chagas Disease

**DOI:** 10.3390/ijms25021202

**Published:** 2024-01-18

**Authors:** Micaela S. Ossowski, Juan Pablo Gallardo, Leticia L. Niborski, Jessica Rodríguez-Durán, Walter J. Lapadula, Maximiliano Juri Ayub, Raúl Chadi, Yolanda Hernandez, Marisa L. Fernandez, Mariana Potenza, Karina A. Gómez

**Affiliations:** 1Instituto de Investigaciones en Ingeniería Genética y Biología Molecular “Dr. Héctor N. Torres” (INGEBI-CONICET), Buenos Aires 1428, Argentina; micaelaossowski@hotmail.com (M.S.O.); jpgallardo2013@outlook.com (J.P.G.); niborski.leticia@gmail.com (L.L.N.); jess.rodridu@gmail.com (J.R.-D.); 2Instituto Multidisciplinario de Investigaciones Biológicas de San Luis (IMIBIO-SL-CONICET), Facultad de Química Bioquímica y Farmacia, Universidad Nacional de San Luis, San Luis 5700, Argentina; wlapadula@gmail.com (W.J.L.); mjuriayub@hotmail.com (M.J.A.); 3Hospital General de Agudos “Dr. Ignacio Pirovano”, Buenos Aires 1430, Argentina; raulchadi@hotmail.com; 4Instituto Nacional de Parasitología “Dr. Mario Fatala Chaben”, Buenos Aires 1063, Argentina; yoy115@hotmail.com (Y.H.); marisa.fernandez@gmail.com (M.L.F.)

**Keywords:** Chagas disease, *Trypanosoma cruzi*, *Leishmania* spp., diagnosis, novel antigen, quinoproteins, Tc323

## Abstract

Chagas disease is caused by the parasite *Trypanosoma cruzi*. In humans, it evolves into a chronic disease, eventually resulting in cardiac, digestive, and/or neurological disorders. In the present study, we characterized a novel *T. cruzi* antigen named Tc323 (TcCLB.504087.20), recognized by a single-chain monoclonal antibody (scFv 6B6) isolated from the B cells of patients with cardiomyopathy related to chronic Chagas disease. Tc323, a ~323 kDa protein, is an uncharacterized protein showing putative quinoprotein alcohol dehydrogenase-like domains. A computational molecular docking study revealed that the scFv 6B6 binds to an internal domain of Tc323. Immunofluorescence microscopy and Western Blot showed that Tc323 is expressed in the main developmental forms of *T. cruzi*, localized intracellularly and exhibiting a membrane-associated pattern. According to phylogenetic analysis, Tc323 is highly conserved throughout evolution in all the lineages of *T. cruzi* so far identified, but it is absent in *Leishmania* spp. and *Trypanosoma brucei*. Most interestingly, only plasma samples from patients infected with *T. cruzi* and those with mixed infection with *Leishmania* spp. reacted against Tc323. Collectively, our findings demonstrate that Tc323 is a promising candidate for the differential serodiagnosis of chronic Chagas disease in areas where *T. cruzi* and *Leishmania* spp. infections coexist.

## 1. Introduction

Neglected diseases caused by trypanosomatids constitute severe health and socio-economic concerns, especially in tropical and low/middle-income countries [1]. Among them, Chagas disease is caused by infection with the hemoflagellate parasite *Trypanosoma cruzi.* Its transmission to humans includes contamination with triatomine vector feces/urine, blood transfusions, or even oral and transplacental routes [2,3,4]. Migratory movements have transformed this disease from a Latin American problem to a global one, with over 7 million people worldwide estimated to be infected by *T. cruzi* [4,5,6]. After the initial infection with the parasite, the disease develops progressively in two phases. The acute phase is short and exhibits high parasitemia. Clinical manifestations are rare and mostly exhibit non-specific symptoms such as fever, diarrhea, enlarged lymph glands, anorexia, headache, swelling, and abdominal or chest pain [7]. If no treatment with trypanocidal drugs (benznidazole or nifurtimox) is provided, the infection becomes chronic. Up to 70% of people with chronic Chagas disease (CCD) experience the infection without any detectable clinical alteration, while the rest progress to mild, moderate, or severe cardiac, digestive, neurological, or mixed disorders, with a higher risk of sudden death [6,7,8].

The diagnosis varies according to the acute or chronic stages of the disease. During the former phase, the detection of the parasite in blood samples via a microscopic examination is a highly touted proceeding. On the contrary, in the latter form, the diagnosis is essentially serological due to low parasitemia and high levels of antibodies against *T. cruzi* antigens [9,10]. Given the absence of a reference or so-called “gold standard” test, the current recommendation is the use of at least two assays based on different antigen preparations for reliable diagnosis [11]. Hence, the detection of *T. cruzi*-specific IgGs is achieved using enzyme-linked immunoassays (ELISA), indirect immunofluorescence, and indirect hemagglutination assays (IHAs). The performance of these methods is affected by the antigenic cross-reactivity with blood samples from patients infected with other trypanosomatids that share mammalian hosts, particularly with *Leishmania* spp. [12,13,14]. It is well-documented that the close phylogenetic relationships between both parasites and the overlap in the endemic areas in Latin America exist [13,15,16,17,18,19,20]. Furthermore, another important aspect is the presence of different antigenic features of *T. cruzi* parasite strains, which are also heterogeneously distributed in different geographical regions and lead to widely diverse humoral immune responses in the host [21].

With the intention to deeply investigate the anti-*T. cruzi* antibody repertoire, our laboratory has performed phage display libraries derived from the B-cell mRNA of patients with CCD and cardiac disorders, constructed using either Fab or single-chain variable fragment (scFv) formats [22,23]. One of the scFv libraries screened against the *T. cruzi* epimastigote parasite lysate allowed the isolation of different monoclonal antibodies, each one with its own binding capacity and protein specificity [24]. In recent work, we demonstrated that the scFv A2R1 recognizes tubulin expressed in all forms of *T. cruzi* as well as in other members of the kinetoplastid family, and this antigen–antibody interaction depends on any post-translational modifications contained in the protein. Interestingly, we also showed that scFv A2R1 binds to mammal tubulin present in the peripheral and central nervous system, supporting the notion that an autoreactive phenomenon occurs in chronically infected patients, and this could be linked to the pathophysiology of Chagas disease [24].

In this study, we identified the hypothetical protein of 323 KDa, herein named Tc323, as the target of another monoclonal antibody isolated from the human single-chain variable fragment phage-display library, scFv 6B6. We found that this protein is exclusively present in different strains and developmental stages of *T. cruzi* but is absent in mammals and other pathogen members of the Trypanosomatidae family. Interestingly, Tc323 has been recognized using the plasma from patients with chronic Chagas disease, while samples from individuals infected with *Leishmania* were not reactive, spotlighting this protein as a possible novel marker for the diagnosis, progression, and treatment efficacy assessment of *T. cruzi* infection.

## 2. Results

### 2.1. The scFv 6B6 Specifically Recognizes a Protein from T. cruzi but Not from Related Pathogens

As mentioned above, scFv 6B6 was isolated from antibody phage-display libraries constructed from the B cell cDNA of CCD subjects and screened against the *T. cruzi* parasite lysate coated on an immunotube [24,25]. The antibody was then expressed and purified from *E. coli*. A Western Blot (WB) assay revealed that scFv 6B6 recognized a protein of ~315 kDa in lysates from different strains and life stages of *T. cruzi*, while no bands were observed in the insect stage extracts of the phylogenetically related pathogens *T. brucei* or *L. mexicana* (Figure 1A,B). A similar lack of reactivity was observed when employing the scFv 6B6 against protein extracts derived from bloodstream forms of *T. brucei* and various *Leishmania* species (Appendix A). Of note, we observed a tendency for a higher expression level of Tc323 in the epimastigote stage when compared to the trypomastigote and the amastigote forms. In the same line, lysates from Dm28c (TcI) and HE (TcV) strains exhibited an upward variation in the expression level of the target protein compared to the other three strains tested. The results were confirmed using the ELISA assay in which scFv 6B6 displayed preferential binding to samples from *T. cruzi*, while no significant reaction was observed against *T. brucei* or *L. mexicana* (Figure 1C). In addition, no reactivity was detected against mammalian protein extracts (Appendix A). In summary, these findings further suggest that the protein recognized by scFv 6B6 is exclusively present in *T. cruzi*.

### 2.2. Tc323 Is the Target Antigen of scFv 6B6

Further, we sought to identify the protein recognized by scFv 6B6 using immunoprecipitation (IP) followed by mass spectrometry analysis (MS). As the full-length antibody results in increased sensitivity in IP experiments compared with its fragments [26], we first generated a chimeric 6B6 (chim m6B6) antibody containing human scFv 6B6 variable domains and the murine IgG1k constant domains. After confirming its specificity through WB analysis (Appendix A), chim m6B6 was covalently bound to a protein A/G agarose resin to immunoprecipitate the target antigen from a *T. cruzi* lysate. Next to the IP, eluted proteins were resolved using SDS-PAGE gel electrophoresis for either Coomassie blue staining or WB analysis. The visualization of proteins in the gel revealed the following two major bands: one at the expected molecular weight of ~315 kDa and another at ~50 kDa. Only the upper protein band was specifically recognized by the scFv 6B6 antibody in the WB (Figure 2A). These results suggest that the ~315 and ~50 kDa bands corresponded to the *T. cruzi* antigen and the heavy chain of the chim m6B6 antibody, respectively.

Then, the bands were excised from the gel and analyzed using LC-MS/MS after trypsin proteolytic digestion. The database search results for the peptide mass fingerprints obtained from MS/MS spectra allowed the identification of the chim m6B6-binding antigen. A total of 14 peptides corresponding to the main hit, with a coverage of 8%, showed an identity with TcCLB.504087.20, a *T. cruzi* protein with a predicted molecular weight of 323 kDa, hereinafter referred to as Tc323 (Figure 2B,C).

Next, we performed homology searches on the Kinetoplastida genomic database using BLASTp, tBLASTn, and PSI-BLAST. As shown in Figure 2D, the presence of Tc323 homologs is restricted to a few species of the *Trypanosoma* genus, including all the sequenced strains of *T. cruzi*, some non-pathogenic trypanosomes and *Paratrypanosoma confusum* (blue branches) (Text S1 alignment). No hits were retrieved from any species of *Leishmania* (including *L. major*, *L. amanzonensis*, *L. braziliensis*, *L. infantum*) and *Trypanosoma brucei* genomes (blue branches), explaining the absence of reactivity of the 6B6 antibody against the protein extracts of these species.

The presence of Tc323 homologs in representative *T. cruzi* lineages and its absence in *T. brucei* was confirmed via synteny analyses (Appendix A). Moreover, we confirmed that there is a single copy of Tc323 per haploid genome in *T. cruzi*. Indeed, this gene is located on chromosomes 24 and 23 of the fully assembled genomes from Y (GCA_015033655) and Brazil (GCA_015033625) strains, respectively.

A phylogenetic tree built with the nucleotide sequences encoding the Tc323 Trypanosomatidae homologous retrieved revealed a very low variability among *T. cruzi* lineages (see short branches in Figure 2E). As depicted in Appendix A, nucleotide sequence identities among Tc323 from *T. cruzi* lineages were higher than 97%.

### 2.3. Tc323 Localizes Intracellularly and Exhibits a Membrane-Associated Pattern

The antigen Tc323 is annotated as a hypothetical protein at TriTrypDB and exhibits putative quinoprotein alcohol dehydrogenase (ADH)-like structural motifs (IPR011047) without previous functional studies. Sequence analysis using InterProScan predicted the presence of an N-terminal signal peptide, a central non-cytoplasmic extracellular domain, a transmembrane motif, and a short C-terminal cytoplasmic region (Figure 3A).

In order to determine the sub-cellular localization of Tc323 in the main developmental forms of *T. cruzi*, we performed WB, immunofluorescence microscopy, and the flow cytometry of epimastigotes, amastigotes, and trypomastigotes, using scFv 6B6 and chim m6B6 antibodies. WB analysis in Figure 3B showed that Tc323 was present in either sample-containing soluble or membrane-enriched fractions of *T. cruzi* cell extracts, contrary to the signal observed for cyclophilin TcCyp19, a cytoplasmic protein used as fraction-enrichment control [27]. The anti-BiP antibody recognizes the Endoplasmic Reticulum-resident chaperone BiP [28]. This protein is localized in the lumen of the ER and associated with intact microsomal vesicles [29], serving as a loading control when comparing the soluble or membrane-associated presence of Tc323 (Figure 3B). The immunofluorescence microscopy of *T. cruzi* cells, with fixed-permeabilized or unpermeabilized Triton X-100, revealed that Tc323 is mainly located intracellularly in the three developmental forms (Figure 3C). Notably, the signal detected using the scFv 6B6 antibody exhibits a different pattern compared to that detected by the anti-TcCyp19 antibody, suggesting a non-cytosolic distribution of Tc323 inside the cell. However, the localization pattern of Tc323 did not appear to be associated with Endoplasmic Reticulum or Golgi membranes (Appendix A). In addition, the specificity of scFv 6B6 was also investigated using a recombinant antibody scFv A2R1, which carried the same Fc fragment but a Fab region recognizing tubulin from *T. cruzi*. The immunostaining microscopy images showed that these antibodies exhibit different localization patterns (Appendix A).

The subcellular localization was also investigated by studying the reactivity of the chim m6B6 antibody against live or permeabilized *T. cruzi* parasites via flow cytometry (Figure 3D). The rightward shift of the histogram was observed only when the parasites were permeabilized, further indicating that Tc323 was more likely localized within the cell than on the plasma cell surface membrane.

### 2.4. Tc323 Is Immunogenic

After identifying the antigen that bound to scFv 6B6 and studying its presence within the trypanosomatid parasites, we moved forward to investigate the antigenic properties of Tc323.

First, we simulated the native antibody–antigen interaction using the online tool HADDOCK to identify the binding sites between scFv 6B6 and Tc323. For this, the predicted structures of both proteins were built with RaptorX and validated by PROCHECK programs. Only the Tc323 and scFv 6B6 structural models that exhibited the highest number of residues within the favored regions of the Ramachandran plot (80.4 and 85.5%, respectively) were selected and submitted to the HADDOCK server for docking. As a result, two predicted interaction sites between scFv 6B6 and Tc323 (clusters) were identified with 2% of the water-refined models. One cluster exhibited positive HADDOCK and Z-scores and was discarded, while the other presented negative values for these parameters and was selected for further analysis. The HADDOCK score was calculated as a combination of van der Waals, electrostatic, empirical desolvation, and restraint energies together with the buried surface area (bsa). The overall model quality Z-score is represented by the number of standard deviations from the average to a particular cluster. In addition, the bsa parameter, which measures the amount of the protein surface not in contact with water, exhibited a positive high value for this cluster, indicating that the structural complex is compact. The best structural complex from this cluster was downloaded as a PDB file and visualized using PyMOL software v2.5.7. As shown in Figure 4A, residues A165, D204, E223, F193, G212, K229, R186, S189, and S214 from scFv 6B6 and G2252, G2307, K2262, N2266, N2183, N2271, P2275, Q2104, and S2144 of Tc323 were identified as responsible of the antibody–antigen interaction, respectively. Considering that a linear epitope is defined by a short continuous sequence of amino acids [30], our in silico analysis proposes that the antibody binding site of Tc323 is a discontinuous or conformational region.

Then, we wondered if Tc323 had a potential utility in the serological diagnosis of CCD. For this purpose, we selected plasma from patients with different clinical manifestations of CCD, cutaneous leishmaniasis (CL), mixed-infection (MI), and non-infected subjects (NI) as a negative control to assess their reactivity against Tc323 (Appendix A).

We first measured the anti-*T. cruzi* and anti-*L. mexicana* antibody level using an *in-house* ELISA (Appendix A). In line with previous findings [15,31,32,33,34], our results showed that the level of specific anti-*T. cruzi* antibodies ranged equally in both groups of patients with CCD, whereas the titer of specific antibodies in subjects with leishmaniasis was low. Notably, these samples displayed a preferential binding to *T. cruzi* (Appendix A). When we measured plasma reactivity via WB using the immunoprecipitated Tc323, we found that the protein was only recognized by plasma from patients with CCD and MI (Figure 4B–D). No binding was observed with plasma from patients with CL and from non-infected subjects (Figure 4B), except in one sample that showed a recognition of the protein via immunoblotting (Figure 4C,D).

## 3. Discussion

In this work, we detail the characterization of a monoclonal antibody obtained from a human single-chain Fv phage display library, along with the identification of its target antigen in *T. cruzi*. The isolated antibody binds to Tc323, an uncharacterized hypothetical protein with higher abundance in the epimastigote stage compared to the other two developmental forms present in mammals. Based on these data, we speculate that the developmentally regulated expression of Tc323 could reflect its relevant role in the survival of the parasite in the invertebrate host. Our results also demonstrate some variations in the Tc323 protein level among the *T. cruzi* lineages tested (TcI, TcII, TcV, and TcVI). In accordance, parasite strains display significant variability in their gene expression patterns in response to environmental adaptation [35]. The differential expression of proteins between TcI (PR150) and TcII (452) genotypes isolated from chronic Chagas disease cardiac patients has been well documented. Using 2D gel electrophoretic and mass spectrometry analysis, the authors demonstrated the overrepresentation of proteins related to stress response and amino acid and lipid metabolisms in strain 452. However, high levels of proteins linked to central metabolic pathways characterized the sample of the TcI-representative genotype. These findings could potentially explain the biological behavior of *T. cruzi* observed in in vitro and in vivo studies [36,37].

In addition, we found that Tc323 localizes in the intracellular space of the parasite, exhibiting a membrane-bound and associated pattern. In line with this finding, proteomic studies identified peptides of Tc323 in intracellular vesicles of *T. cruzi*, like reservosomes or the contractile vacuole [38,39]. In our work, no signal of Tc323 was detected using microscopy and flow cytometry on the surface of non-permeabilized parasites, as opposed to another proteomic study where peptides of this protein were identified in the cell surface of epimastigotes [40], and the in silico predicted extracellular localization. The discrepancy regarding the protein localization on the cell’s surface might be the result of the intrinsic limitations and/or different methodological sensitivity of these research tools that need further investigation. The presence of a consensus sequence for a signal peptide at the *N*-terminus, its localization pattern, and the identification of this protein in the secretome of *T. cruzi* [41] allows us to hypothesize that this protein could be transported through the secretory pathway of the parasite. As a result, Tc323 could be secreted to the extracellular space, promoting a humoral response during infection.

Interestingly, Tc323 exhibits structural motifs belonging to the quinoprotein alcohol dehydrogenase (ADH)-like superfamily, a family of enzymes that require the pyrroloquinoline quinone (PQQ) as a cofactor to catalyze the oxidation of alcohols [42,43]. The quinoprotein ADH-like superfamily includes proteins that might not necessarily have the exact same catalytic function but share similar structural characteristics and sequence motifs. These proteins are often involved in various redox reactions, participating in the transfer of electrons and the oxidation or reduction of substrates. The superfamily encompasses a diverse range of enzymes and plays a role in various biological processes [44,45]. To our knowledge, there is no PQQ-dependent ADH or any other enzymatic activity reported in trypanosomes to date. The ADH gene described to date in *T. cruzi* is the NAD(P)-dependent enzyme of 42 kDa, located in the mitochondria and exhibiting a higher level of similarity to prokaryotic ADHs than to orthologs identified in *Leishmania* species. In line with evolutionary adaptation, the abundance of TcADH varies in benznidazole-susceptible or -resistant strains of *T. cruzi* [46]. In fact, TcADH is overexpressed in naturally benznidazole-resistant strains compared to drug-sensitive ones [47]. Thus, the differential expression of certain proteins and enzymatic activities may promote the establishment of infection and/or determine drug resistance.

It is worth mentioning that Tc323 lacks homologs in other eukaryotes out of kinetoplastids, including the mammalian hosts that *T. cruzi* parasitizes. The presence of homologs to Tc323 is only restricted to five species of trypanosomes from stercorarian clades, as well as *T. rangeli* and the early-branching *Paratrypanosoma*. It was previously reported that *Paratrypanosoma* and stercorarian trypanosomes share more ancestral genes in comparison to salivarian clades, which include *T. brucei* and some *Leishmania* [48,49]. In line with the evolutionary history of Tc323, scFv 6B6 showed no reactivity against the total protein extracts from *T. brucei,* which are different species of *Leishmania* and mammals. Hence, we decided to explore the antigenicity of Tc323 as a diagnostic marker capable of distinguishing between infection of *T. cruzi* and *Leishmania* spp.

A crucial factor in obtaining reliable diagnoses in areas where infections by *T. cruzi* and *Leishmania* spp. occur concomitantly is to assess the absence of cross-reactivity between the antigens of these parasites [14,50,51]. The false-positive results in individuals exposed to *Leishmania* spp are not only demarcated to serological tests employing the parasite lysate but also to those based on recombinant antigen preparations [10,13,18,51]. Here, we showed that only plasmas from patients infected with *T. cruzi* or co-infected with *Leishmania* spp. reacted to immunoprecipitated Tc323 on WB assays. No reactivity was found with samples from patients affected with leishmaniasis. Although 1 out of 10 plasmas from non-infected individuals seemed to recognize Tc323, we could not rule out the possibility that this subject was misdiagnosed or that the sensitivity of the WB assay using the immunoprecipitated Tc323 was not good enough and delivered a false positive. Another key factor that affects the efficacy of available serological tests is the genetic diversity of *T. cruzi* strains, which, in turn, defines the antigenic constitution of the parasite [52,53]. Interestingly, we used Tc323-positive samples derived from Argentina, Bolivia, and Paraguay, where the genotypes TcV/TcVI, TcV, and TcIII/TcV, respectively, are most frequent [54]. It is our intention to draw attention to this point because the performance of the serological diagnostic assays depends on the geographic origin of plasma/serum samples.

Finally, docking simulations predicted the presence of a conformational epitope, rather than a linear one, recognized by scFv 6B6 in the putative tridimensional structure of Tc323. Our ongoing studies are focused on obtaining the recombinant Tc323 protein and/or its epitope-containing fragment that is suitable for use in ELISA assays. In accordance and keeping in mind our previous results in which scFv A2R1 bound to its target, tubulin, only when it underwent any eukaryotic-specific post-translational modification, we not only restricted our production to prokaryotic expression systems but also considered *L. tarentolae* as an expression host.

## 4. Materials and Methods

### 4.1. Production of scFv 6B6 and Chim m6B6 Antibodies

*E. coli* HB2151 cells transformed with the pUC119-VSV-HIS6-6B6 plasmid were grown at 37 °C in a 2x tryptone/yeast medium (2xTY) supplemented with 2% (*w*/*v*) glucose and 100 μg/mL ampicillin until OD_600nm_ = 0.6. The expression of the recombinant antibody was performed by growing the culture in a 2xTY-Ampicillin medium lacking glucose plus 1 mM isopropyl-β-D-thiogalactopyranoside (IPTG) for 18 h at 20 °C. The cells were collected via centrifugation, resuspended in 0.2 M Tris-HCl pH 8.0, 0.5 M EDTA, 0.5 M sucrose, protease inhibitor cocktail (Roche Applied Sciences, Mannheim, Germany), and then incubated on ice for 30 min. The periplasmic fraction containing scFv 6B6 was separated from the cellular debris via centrifugation and dialyzed against 0.1 M imidazole, 0.05 M Na_2_HPO_4_, 0.3 M NaCl pH 8.0 for 16 h at 4 °C. The scFv 6B6 was purified by affinity chromatography using a nickel-charged agarose resin (QIAGEN, Venlo, The Netherlands) according to the manufacturer’s instructions. The scFv 6B6 concentration was calculated using spectrometry (ε_280nm_ = 1.55 M^−1^ cm^−1^).

The mouse-human chimeric antibody 6B6 (chim m6B6) was assembled by fusing the antigen-binding region of scFv 6B6 (variable domains of the heavy and light chains, VH and VL, respectively) with mouse C gene segments [55]. Concisely, VH and VL regions were amplified via PCR following cloning into eukaryotic expression vector-containing mouse genomic IgG1 heavy and kappa light chain constant regions and regulatory elements of the immunoglobulin locus. The expression of chim m6B6 was performed in HEK293 cells. The secreted chim m6B6 antibody was purified from the cell culture supernatants via Protein A chromatography (MabSelect SuRe™, GE Healthcare, Little Chalfont, UK).

The expression and reactivity of both antibodies were determined using Western Blot (WB).

### 4.2. Immunoprecipitation and Nano-LC Mass Spectrometry

Immunoprecipitation (IP) experiments were performed using a commercial kit (Thermo Pierce Scientific, Carlsbad, CA, USA) following the manufacturer’s instructions. Briefly, 1 mg of *T. cruzi* epimastigote lysate (CL-Brener) was passed through the protein A/G agarose column to minimize the non-specific binding of the parasite proteins to the resin. In parallel, 20 µg of chim m6B6 was covalently cross-linked to the protein A/G agarose using DSS (Thermo Pierce Scientific, CA, USA). 1 h at room temperature (RT). Then, the agarose-coupled chim m6B6 was incubated with the pre-cleared *T. cruzi* lysate under gentle shaking for 1 h at 4 °C. After washing, the bound proteins were eluted with 0.1 M of glycine-HCl pH 2.0 and immediately neutralized with 5 µL of 1 M Tris-HCl pH 9.0 to avoid protein degradation.

The eluted fractions were resuspended in a 5X sample buffer (0.3M Tris-HCl, 5% (*w*/*v*) SDS, 50% (*v*/*v*) glycerol, 0.05% (*w*/*v*) pink lane marker tracking dye; pH 6.8) and subjected to SDS-PAGE. After staining with colloidal Coomassie Blue G-250, the protein bands were excised and analyzed with LC-MS/MS at FingerPrints Proteomic Facility, University of Dundee, Scotland. The proteins were identified by searching peptide mass fingerprints against TriTrypDB and UniProt databases [56,57] and using the MASCOT search engine (www.matrixscience.com (accessed on 16 June 2018)).

### 4.3. In Silico Predictions and Phylogeny Analyses

For the phylogenetic analysis of Tc323, an evolutionary history of the protein was constructed using the PhyML software (v3.0) [58]. For this analysis, the sequences were obtained through BLASTn searches within the TriTrypDB database [56] and NCBI (www.ncbi.nlm.nih.gov (accessed on 14 July 2023)) using the nucleotide sequence of Tc323 as a query. The absence of homologous sequences in other Kinetoplastea species was confirmed through BLASTp, PSI-BLAST, and tBlastn searches using all Tc323 homologs as queries. Multiple sequence alignment was performed with the MAFFT program [59], and the output visualization and editing were carried out with MEGA software (version X, Penn State, Philadelphia, PA, USA) [60]. Finally, the phylograms were obtained using the maximum likelihood method, with the General Time Reversible (GTR) model according to Smart Model Selection (SMS) [61]. Standard bootstrap analysis was performed with 100 replicates to assess the branch support. The graphical viewer FigTree v1.4.2 (https://tree.bio.ed.ac.uk/software/figtree (accessed on 14 July 2023)) was used to visualize and edit the phylogenetic trees.

Synteny analysis was performed using simple synteny software v1.6 (www.dveltri.com/simplesynteny/ (accessed on 14 July 2023)). A 38.5 kb genomic fragment (nucleotide 42,6366 to 46,4886) of the *T. cruzi* Y strain genome (GenBank CM026649) was used as a reference. The open reading frames present in this region were used to identify synteny regions in Esmeraldo like (GenBank NW001849505), Bug2148 (GenBank NMZN01000123), Brazil (GenBank CM026605), S231 (GenBank OGCJ01000236) *T. cruzi* strains and *T. brucei* TREU927 (GenBank NT165288).

Tc323 protein motifs were predicted using the online tool InterProScan [62]. RaptorX server (http://raptorx.uchicago.edu (accessed on 18 December 2018)) [63] was utilized to predict the tridimensional structural models. The stereochemical quality of the generated protein structures was verified using PROCHECK and the Ramachandran plot [64].

HADDOCK (High Ambiguity-Driven protein–protein DOCKing) v.2.2 web-server [65] was used to carry out flexible docking simulations between the scFv 6B6 antibody and Tc323 protein with default docking parameters (https://wenmr.science.uu.nl (accessed on 4 April 2023)). The docked complex was visualized using PyMOL software (https://pymol.org/2/ (accessed on 4 April 2023)).

### 4.4. Parasite Strains and Growth Conditions

Epimastigote forms of *T. cruzi* from Dm28c, Y, HE, and CL Brener strains, belonging to TcI, TcII, TcV, and TcVI phylogenetic groups, respectively [54], were grown at 28 °C in a liver tryptose infusion medium (LIT) supplemented with 10% (*v*/*v*) heat-inactivated fetal bovine serum (FBS, Natocor, Córdoba, Argentina), 100 U/mL of penicillin, 100 mg/mL of streptomycin and 25 µg/mL of hemin. Cell-derived trypomastigotes from *T. cruzi* the Dm28c strain were obtained from the culture media of infected VERO cells (multiplicity of infection, MOI 2:1) in complete RPMI-1640 medium supplemented with 3% inactivated FBS, 100 U/mL of penicillin, 100 μg/mL of streptomycin, and 2 mM of L-glutamine (all from Gibco, Thermo Pierce Scientific, CA, USA) at 3 days post-infection. Axenic amastigotes were obtained by incubating cell-derived trypomastigotes in RMPI-1640 pH 5.0, supplemented with 0.4% bovine serum albumin (BSA) for 24 h at 37 °C.

Procyclic forms of *T. brucei* and promastigote forms of *L. mexicana* were kindly provided by Dr. Guillermo Alonso (INGEBI, Buenos Aires, Argentina) and Dr. Carlos A. Labriola (Fundación Leloir, Buenos Aires, Argentina), respectively.

### 4.5. Preparation of Cell Extracts

Whole antigenic lysates from different strains and developmental stages of *T. cruzi*, promastigotes of *L. mexicana*, *L. amazonensis*, and *L. infantum*, as well as bloodstream and procyclic forms of *T. brucei* were prepared as previously described [24]. Briefly, parasites were harvested via centrifugation at 1500× *g* for 5 min and washed three times with ice-cold Phosphate Buffer Saline (PBS). Then, the cells were resuspended in a lysis buffer (5 mM MgCl_2_, 25 mM potassium acetate, 1 mM dithiothreitol (DTT), 250 mM sucrose, 1% Triton X-100, 0.1% NP40, 20 mM Tris-HCl and protease inhibitors cocktail), followed by three cycles of freezing-thawing with liquid nitrogen. After lysis, the suspension was filter-sterilized through a 0.2 μm pore size membrane, aliquoted and stored at −80 °C until use.

Cell extracts enriched in soluble and membranous fractions from *T. cruzi* Dm28c epimastigotes, axenic amastigotes, and cell-derived trypomastigotes were prepared as previously reported [29]. In brief, 1 × 10^7^ cells were washed three times with PBS and then suspended in 130 µL of 20 mM Tris-HCl (pH 8.0), 150 mM NaCl, 2 mM MgCl_2_, 2 mM EGTA, and a protease inhibitor cocktail. After freeze and thawing cycles, lysates were centrifuged at 12,000× *g* for 20 min to separate the soluble enriched fraction (supernatant) from the cell pellet. The membranous-enriched fraction was obtained adding 150 µL of the same lysis buffer supplemented with 1% (*v*/*v*) Triton X-100 to the cellular debris.

Organs from 8-week-old C57/BL6 mice were pulverized in liquid nitrogen and minced in a lysing buffer containing 50 mM of Tris-HCl at pH 7.5, 50 mM of NaCl, 1 mM of EDTA, 1 mM of phenylmethylsulfonylfluoride, 1 mM of benzamide, 1% (*v*/*v*) of Triton X-100, 10% glycerol and a protease inhibitor cocktail. After centrifugation at 16,000× *g* for 10 min, protein suspensions were aliquoted and stored at −80 °C. Lysing buffer for liver and muscle tissues contained 250 mM sucrose and 10 mM EDTA.

All sample concentrations were determined using the Bradford assay (BioRad, Hercules, CA, USA).

Unless specified, all procedures were performed at 4 °C.

### 4.6. Enzyme-Linked Immunosorbent Assay (ELISA)

ELISA was performed as previously described [66]. In brief, either 20 µg/mL of parasite protein lysate, 20 µg/mL of mouse organ tissue extracts, or 5 µg/mL of BSA were coated for 16 h at 4 °C in 0.05 M carbonate buffer (pH 9.6) in 96-well plates (NUNC MaxiSorb^TM^, Roskilde, Denmark). After four washes with 1% (*v*/*v*) Tween 20 in PBS (PBST), the plates were blocked for 2 h at RT with 5% (*w*/*v*) of skimmed milk in PBST. Subsequently, plates were incubated with scFv 6B6 antibody (25 μg/mL) or plasma from patients with CCD, cutaneous leishmaniasis (CL), or non-infected (NI) individuals (diluted 1:250–1:100) for 2 h at 37 °C at 30 rpm shaking speeds. After four washes with PBST, the wells were incubated with mouse anti-6×His-HRP antibody (1:3000, clone HIS-1, Sigma Aldrich, St. Louis, MO, USA) or mouse anti-human IgG coupled to Peroxidase (HRP) (1:3000, Sigma-Aldrich, USA) for 1 h at 37 °C. All antibodies and plasma were diluted in 1% (*w*/*v*) skimmed milk in PBST. After incubation with 3,3′,5,5′-tetramethylbenzidina (TMB, Sigma-Aldrich, USA) plus H_2_O_2_ as substrate, the reaction was stopped with 4 N H_2_SO_4_. Optical density (OD) was measured at 450 nm using a VERSAmax^®^ ELISA plate reader (Molecular Devices Corporation, San Jose, CA, USA).

### 4.7. Western Blot

Proteins were separated on 8–12% SDS-polyacrylamide gels and transferred to a nitrocellulose membrane (GE Healthcare, Chicago, IL, USA) via wet electroblotting for 1 h. The membranes were blocked with Tris-buffered saline containing 1% (*v*/*v*) Tween 20 (TBST) and 5% (*w*/*v*) of skimmed milk for 1 h at RT. Subsequently, the membranes were incubated with mouse anti-6×His-HRP (1:3000, clone HIS-1, Sigma-Aldrich, USA), scFv 6B6 (25 μg/mL), chim m6B6 (3 μg/mL), anti-β-tubulin (1:8000, Sigma-Aldrich, USA), anti-TcCyP19 (1:300), and anti-BiP (1:10,000) antibodies or human plasma samples for 1 or 2 h, respectively. After four washings with TBST, the membranes were incubated with rabbit anti-mouse IgG-HRP, mouse anti-Human IgG-HRP (both 1:3000, Sigma-Aldrich, USA), goat anti-mouse IgG-HRP or goat anti-rabbit IgG-HRP (both 1:6000, Calbiochem, USA) for 1 h. All incubations were performed at RT, and antibodies were diluted in 1% (*w*/*v*) BSA or skimmed milk in TBST. Immunoreactive bands were visualized by chemiluminescence (ECL-Plus, Thermo Pierce Scientific, CA, USA) using GeneGnome XRQ (Synoptics Group, Cambridge, UK).

When indicated, densitometric data of the chemiluminescence signal was quantified using ImageJ software v1.54 (National Institutes of Health, Bethesda, MD, USA; https://imagej.nih.gov/ij/ [67]. Data were normalized by determining the relative band intensity (RBI), calculated as the ratio between the plasma sample and the positive control (scFv 6B6) for each experiment. Samples with RBI ≥ 1 were considered positive.

### 4.8. Immunofluorescence Microscopy

Trypomastigote, epimastigotes, and axenic amastigotes of *T. cruzi* (Dm28c) were harvested, washed in PBS, adhered to poly-L-lysine-coated slides, and fixed for 30 min with 4% (*v*/*v*) *p*-formaldehyde in PBS (PFA-PBS). Next, fixed parasites were or were not permeabilized with 0.1% (*v*/*v*) Triton X-100 in PBS for 2 min, blocked with 1% (*w*/*v*) BSA in PBS (BSA-PBS) for 30 min and then incubated with chim m6B6 (13 μg/mL), chim mA2R1 (12 μg/mL), anti-BiP (1:2500), anti-GRASP (1:500) and anti-TcCyp19 (1:300) antibodies. After four washes with PBS, the slides were incubated for 1 h with goat anti-rabbit (H + L) Alexa488 or goat anti-mouse Cy3 (Invitrogen, Thermo Pierce Scientific, CA, USA) secondary antibodies, both diluted 1:500 in BSA-PBS. The slides were washed four times with PBS and mounted with VectaShield (Vector Laboratories Inc., Newark, CA, USA) containing 300 ng/mL 4′,6-diamidino-2-phenylindole (DAPI) for nucleic acid staining. Images were acquired on a Carl Zeiss LSM 880 confocal microscope coupled to the AiryScan module with 63x objective and analyzed using Fiji software v1.52 [68].

### 4.9. Cytometry

*T. cruzi* parasites from the CL strain were incubated with scFv 6B6 (25 µg/mL) and mixed with mouse anti-VSV IgG (1:500, clone P5D4, Sigma-Aldrich, USA) for 1 h on ice. After washing in PBS, parasites were incubated with Alexa488-conjugated goat anti-mouse IgG (1:5000, A-11001 Invitrogen, Thermo Pierce Scientific, CA, USA) diluted in 0.15% gelatin in PBS plus 0.1% sodium azide (PGN) (for 1 h at RT. The procedure described above was performed by fixing parasites with 2% (*v*/*v*) PFA for 1 at RT before (permeabilized) or after (live) incubation with the scFv 6B6 antibody. Data were acquired on a FACScan flow cytometer (Becton Dickinson, Franklin Lakes, NJ, USA) and analyzed with FlowJo software v1.54 (TreeStar Inc., Ashland, OR, USA).

### 4.10. Ethics Statement

The research protocols followed the tenets of the Declaration of Helsinki and were approved by the Medical Ethics Committee of Instituto Nacional de Parasitología “Dr. M. Fatala Chaben” (Study Protocol N° 3-2018 and 19-2019) and Hospital General de Agudos “Dr. I. Pirovano” (Study Protocol N° 56-2015). All enrolled individuals gave written informed consent, according to the guidelines of the ethics committee, before blood collection and after the nature of the study was explained.

### 4.11. Study Population and Human Sample Collection

Samples from patients with CCD were obtained from Instituto Nacional de Parasitología “Dr. Mario Fatala Chaben” and Hospital General de Agudos “Dr. Ignacio Pirovano”. This population had a minimum of two positive serological tests for Chagas disease: indirect immunofluorescence, enzyme-linked immunosorbent assay (ELISA), and/or indirect hemagglutinations (IHAs). The exclusion criteria included at least one of the following conditions: no record of treatment with Benznidazole or Nifurtimox, malignant high hypertension, type-1 diabetes, autoimmune thyroiditis, renal insufficiency, chronic obstructive pulmonary or rheumatic disease, hydroelectrolytic disorders, alcoholism or a history suggesting coronary artery obstruction. The patients who underwent a complete clinical and cardiological examination were classified according to the Kuschnir scale [69] and grouped as follows: G0 (Kuschnir 0 or K0) represents covered individuals with normal ECG and chest radiography; G1 (Kuschnir 1 or K1) indicates individuals with cardiac alterations such as right and/or left branch blockage and different degrees of conductive functional alterations. Ten non-infected subjects were included as the control group (NI).

Blood samples were collected into EDTA-containing tubes (Vacutainer, BD Biosciences, San Jose, CA, USA) and centrifuged at 2000× *g* for 10 min at RT. The supernatant containing plasma samples was aliquoted and stored at −20 °C until use.

Plasma from individuals with chronic gastrointestinal disorders because of *T. cruzi* infection were included in group G1, and those with cutaneous leishmaniasis or mixed infection by both parasites (*T. cruzi* and *Leishmania* spp.) were kindly donated by the Instituto Nacional de Enfermedades Tropicales, Salta, Argentina.

The demographic and clinical characteristics of the study population are summarized in Appendix A.

### 4.12. Statistics

All statistical analyses were conducted using GraphPad Prism v8.0 (GraphPad Software) or R Statistical Software (v4.1 for Windows). The statistical tests are indicated in the Figure legends.

## 5. Conclusions

The characterization of novel proteins from human pathogenic trypanosomatids, such as *T. cruzi*, contributes to unveiling the unknown biological aspects of these parasites [70,71,72] and others. This, in turn, can lead to the discovery of new candidates for diagnoses, prognoses, or biomarkers for treatment cures, as well as a potential drug target [73].

The main limitation of our study was the small size of *T. cruzi* and *Leishmania*-positive samples analyzed. Nonetheless, our results deliver the promise of a novel protein as an antigen with potential utility in the serodiagnosis of chronic Chagas disease. Hence, we are collecting plasma samples from a large cohort of patients from different geographical areas with CDD and other infectious and autoimmune diseases to evaluate the performance (specificity and sensitivity) of this antigen for diagnosis.

Furthermore, Tc323 opens new questions regarding its role in the *T. cruzi* biology and host interaction. In this scenario, the functional characterization of the Tc323 protein is our next step.

## Figures and Tables

**Figure 1 ijms-25-01202-f001:**
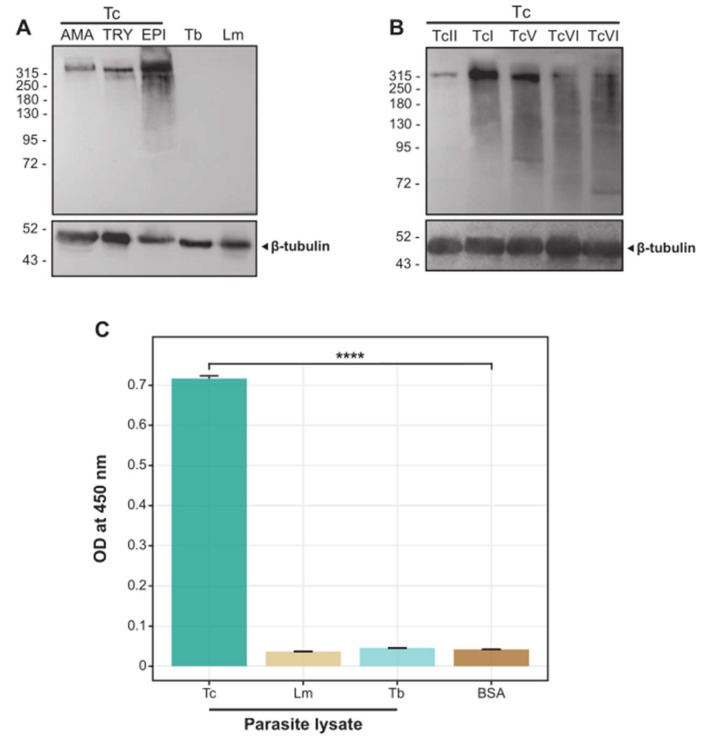
Reactivity of scFv 6B6 against *T. cruzi* and other trypanosomatids using immunoblot and ELISA. (**A**) A total of 30 µg of protein extracts from different parasites and developmental forms were resolved using SDS-PAGE, transferred to a nitrocellulose membrane, and assayed via WB using the antibodies scFv 6B6 (upper panel) and β-tubulin (loading control, lower panel). Tc: *T. cruzi* Dm28c strain. AMA, TRY, and EPI are amastigotes, trypomastigotes and epimastigotes, respectively. Lm: promastigote *L. mexicana*, Tb: procyclic *T. brucei.* Molecular weight markers (in kDa) are indicated on the left. (**B**) A total of 50 µg of protein extracts from epimastigotes of different *T. cruzi* (Tc) lineages were processed as in (**A**). TcI: Dm28c; TcII: Y; TcV: HE, TcVI: CL-Brener; TcVI: Tulahuen. (**C**) ELISA analysis of the relative abundance of the scFv 6B6-binding protein in the total cell extract from *T. cruzi* Dm28c epimastigotes (Tc), *L. mexicana* (Lm) and *T. brucei* (Tb). BSA was used as a control. Data are expressed as the OD_450nm_ (means ± SD; *n* = 2). Differences between the means of binding to different lysates vs. BSA were evaluated using one-way ANOVA followed by Dunn’s multiple comparison tests. **** *p* < 0.0001.

**Figure 2 ijms-25-01202-f002:**
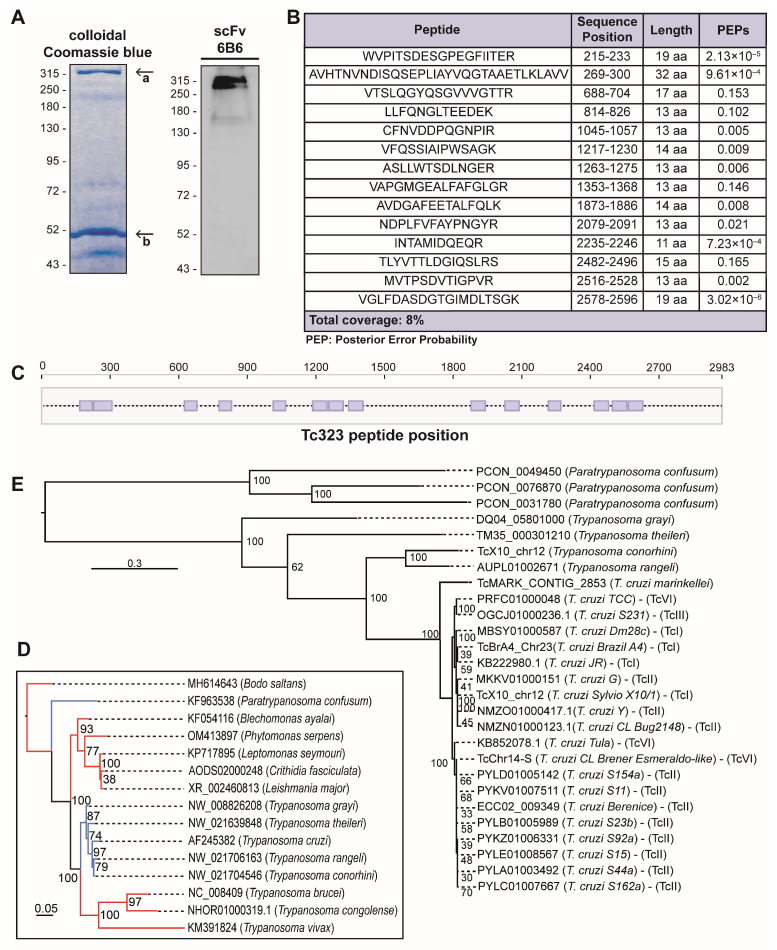
Identification of 6B6 target antigen. (**A**) The *T. cruzi* epimastigote lysate was immunoprecipitated with the chim m6B6 antibody covalently bound to the protein A/G agarose resin. Retained proteins were eluted and resolved with SDS-PAGE (8%), followed by protein staining using colloidal Coomassie Blue (left panel) or transferring to a nitrocellulose membrane and revealing the scFv 6B6 (right panel)**.** The arrows indicate the protein band analyzed using LC-MS/MS (a) and the heavy chain of the chim m6B6 antibody (b). Numbers indicate the molecular weight marker in kDa. (**B**) Deduced amino acid sequences of peptides identified by mass spectrometry. (**C**) The residue position of identified peptides within Tc323 is represented by grey boxes along the protein sequence. (**D**) The maximum likelihood tree of 18S ribosomal RNA was obtained from selected species of the Kinetoplastea class. Bootstrap support values are shown at the nodes. The GenBank numbers belong to the contigs where 18S rRNA is located. Species harboring or lacking Tc323 homologous are indicated with blue and red branches, respectively. Horizontal scale bars represent 0.3 or 0.05 substitutions per position in the nucleic acid sequence, respectively. (**E**) Evolutionary history of Trypanosomatidae Tc323 homologous. Numbers at the branches indicate the maximum likelihood of bootstrap support. The access codes displayed in the tree belong to TriTrypDB or NCBI.

**Figure 3 ijms-25-01202-f003:**
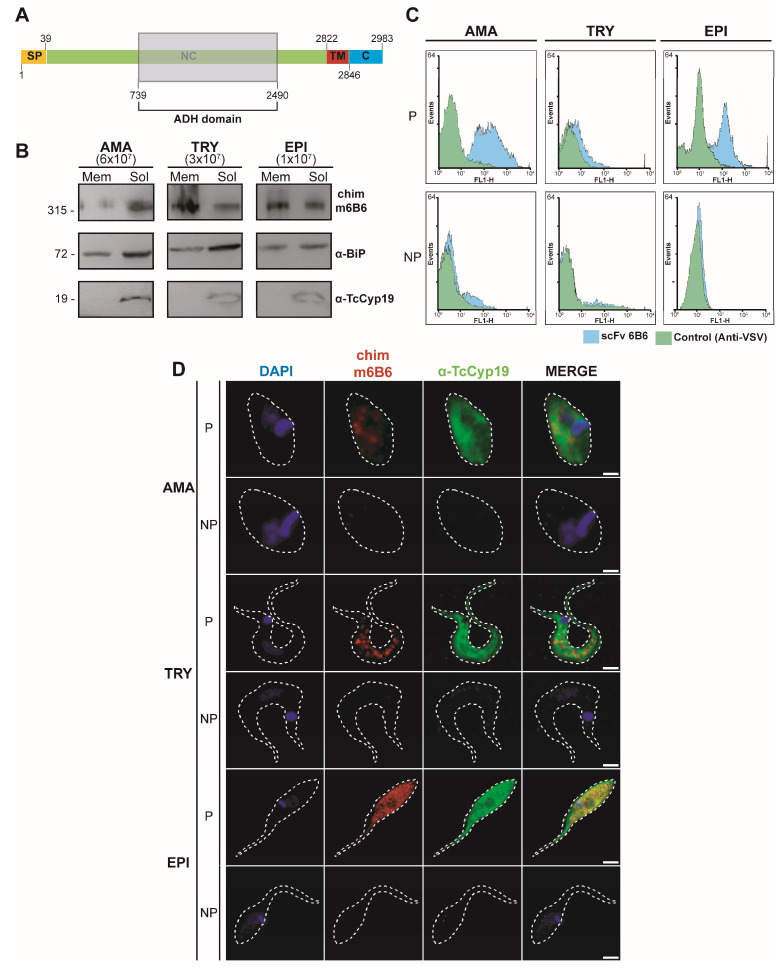
Subcellular localization of Tc323 in *T. cruzi*. (**A**) Schematic representation of Tc323. The signal peptide (PS), non-cytosolic (NC), transmembrane (TM), and cytosolic (C) regions predicted on the InterProScan server are indicated. The region with structural homology to Quinoprotein ADH is indicated by the shaded box. (**B**) The WB of membrane-enriched (Mem) and soluble (Sol) fractions of different *T. cruzi* forms (Dm28c) revealed chim m6B6, anti-BiP and anti-TcCyp19 antibodies. Numbers in brackets indicate the number of parasites used to obtain Mem and Sol fraction samples. On the left, numbers represent the molecular weight marker in kDa. (**C**) The binding of chim m6B6 6B6 to live or fixed-permeabilized different developmental forms of *T. cruzi* from the CL strain via flow cytometry. Histograms depict cell counts (*y*-axis) vs. fluorescence intensity (*x*-axis). Samples incubated with scFv 6B6 are shown as sky blue-filled histograms, while the samples corresponding to the secondary antibody, anti-VSV (negative control), are depicted as green-filled histograms. In (**B**–**D**): AMA, TRY and EPI indicate amastigote, trypomastigote and epimastigote forms, respectively. (**D**) The immunofluorescence microscopy of permeabilized (P) or non-permeabilized (NP) *T. cruzi* developmental forms using chim m6B6 (red) and anti-TcCyp19 (green) antibodies. Nuclei and kinetoplasts were stained with DAPI (blue). Bars: 1 μm for AMA, 2 μm for TRY, 3 μm for EPI.

**Figure 4 ijms-25-01202-f004:**
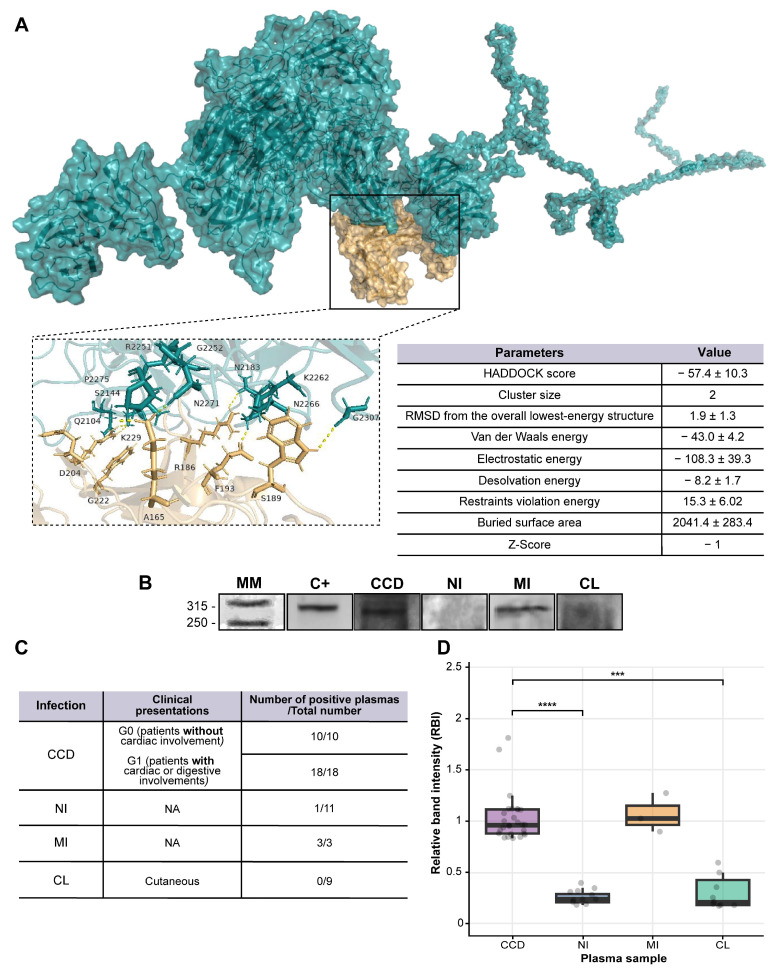
The immunogenic potential of Tc323. (**A**) Molecular docking analysis. The table shows the score values obtained for the simulation of the Tc323-scFv 6B6 interaction using the HADDOCK program (right panel). Protein–protein complex showing surface interactions between Tc323 (blue) with scFv 6B6 (light brown) visualized using PyMOL software. The main residues involved in the ligand–protein complexes are represented as a ribbon diagram in a boxed zoom-in view showing the protein–protein interaction region (left panel). (**B**) Representative WB shows the reactivity of different plasma samples against Tc323. The protein was immunoprecipitated with chim m6B6, resolved onto SDS-PAGE, transferred to nitrocellulose membranes, and revealed with the scFv 6B6 antibody (C+: positive control) or plasmas from non-infected (NI), chronic Chagas disease (CCD), mixed (MI) or cutaneous leishmaniasis (CL)-infected individuals. MM: molecular weight marker in kDa. (**C**) Plasma reactivity against Tc323 for the whole cohort of individuals. The number of individuals and clinical manifestations of the study population are indicated. NA: not applicable. (**D**) Densitometric analysis of the recognition of Tc323 by plasma patients described in (**C**) using ImageJ. Statistical analysis was performed using the non-parametric Kruskal–Wallis test, followed by Dunn’s multiple comparison tests. *** *p* < 0.001; **** *p* < 0.0001.

## Data Availability

Data are contained within the article and Appendix A.

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
