# Peer review of "Characterization of Novel Trypanosoma cruzi-Specific Antigen with Potential Use in the Diagnosis of Chagas Disease"

_ijms, 2024, doi:10.3390/ijms25021202_

Round 1

Reviewer 1 Report

Comments and Suggestions for Authors

The manuscript entitled "Characterization of novel Trypanosoma cruzi specific antigen with potential use in the diagnosis of Chagas Disease" Title, abstract and overall rationale of work is written satisfactory. Still, there are major concerns, which needs to be addressed before publication.

1) Abstract is written well and descriptive. However, the some important keywords are missing like Tc323 and others. I suggest author please incorporate this important keywords.

2) Introduction is written rationally however, author need add some more paragraph about the Tc323 and also introduce about this Tc323 more details.

3) In the result section some important part are missing for example functional characterization of the Tc323 protein and their validation. This is important for this Tc323 protein validation.

4) Material methods section is written details

5) Discussion section: This section author need to improve because author written the results part but they do not discuss properly and I saw the lack of discussion in this manuscript. I recommend author, they should elaborate the discussion part and author need to revise and compare the study with relevant study.

6) Conclusion: Author need to include conclusion section and author need to limitation of study and future prospective.

Comments on the Quality of English Language

English quality is good.

Author Response

Response to Reviewers' Comments

We express our gratitude to the editors and reviewers for their thorough examination of the manuscript and their constructive feedback. We have carefully considered their comments to improve and clarify the manuscript. Below, you will find a comprehensive point-by-point response to all the comments. All page numbers refer to the revised manuscript file with tracked changes.

Reviewer #1

  1. Abstract is written well and descriptive. However, the some important keywords are missing like Tc323 and others. I suggest author please incorporate this important keywords.

Response: We have included Tc323 protein as a keyword in accordance with the Reviewer's recommendation.

  1. Introduction is written rationally however, author need add some more paragraph about the Tc323 and also introduce about this Tc323 more details.

Response: We have modified the text by including more details about Tc323 protein [Lines 83-91].

  1. In the result section some important part are missing for example functional characterization of the Tc323 protein and their validation. This is important for this Tc323 protein validation.

Response: As it was mentioned in the current version of the manuscript [Lines 201-206], we are addressing the functional characterization of Tc323 with various experimental approaches. Our commitment involves overexpressing the PQQ-dependent alcohol dehydrogenase-like domain of Tc323 using different expression platforms to assess its enzymatic activity and identify the potential substrate(s). Additionally, we are employing CRISPR/Cas9-based techniques to generate T. cruzi parasites lacking the expression of Tc323. This aims to explore the parasite´s fitness in proliferation, differentiation, and invasion of mammalian host cells. Furthermore, we are in the process of designing PCR-based gene synthesis to build the entire sequence of Tc323 and overexpress this protein in T. cruzi, enabling the study of its phenotype and physiology.

Moreover, we are conducting endogenous tagging of Tc323 to trace its localization along the secretory pathway of the parasite. In the near future, we expect to share the conclusive results of these approaches and any other significant findings with the scientific community, uncovering the Tc323 function in the T. cruzi biology.

  1. Material methods section is written details

Response: Thanks for your assessment.

  1. Discussion section: This section author need to improve because author written the results part but they do not discuss properly and I saw the lack of discussion in this manuscript. I recommend author, they should elaborate the discussion part and author need to revise and compare the study with relevant study.

Response: Thank you for pointing this out. It has been developed accordingly, please notice all the numerous changes made in the revised manuscript. 

  1. Conclusion: Author need to include conclusion section and author need to limitation of study and future prospective.

Response: As Reviewer´s suggestion, we have added the Conclusion section, encompassing the study´s limitations and future prospective.

Reviewer 2 Report

Comments and Suggestions for Authors

In this paper, Ossowski et al. characterize a novel Trypanosoma cruzi antigen Tc323, that is recognized by a scFv (6B6) previously identified via screening of a scFv library derived from chronic Chagas disease patients with T. cruzi parasite lysate. The authors recombinantly express 6B6 in E. coli and use it to probe Western blots of T. cruzi lysate, which shows that it recognizes a ~315kDa protein that is expressed in all major T. cruzi parasite developmental stages and in different T. cruzi strains, but not in related T. brucei and Leishmania parasites, or in mammalian cells. The identity of the 6B6 scFv target antigen was determined via reconstitution of 6B6 into a chimeric antibody which was used for immunoprecipitation followed by mass spectrometry. Sequence database searching revealed that Tc323 is highly conserved among T. cruzi strains and that T. brucei or Leishmania species appear to lack homologues. A combination of cell fractionation, immunofluorescence and flow cytometry using permeabilized and non-permeabilized T. cruzi cells of different life-cycle stages was used to show that Tc323 is expressed intracellularly. The authors also use molecular docking to predict that 6B6 recognizes a conformational epitope in Tc323. Finally, and importantly, the authors use patient serum samples to show that Tc323 is a promising target for serodiagnosis of T. cruzi infection, including mixed infections with Leishmania. The conclusions are generally well-supported and the work well-described. However, I have some comments that need to be addressed before publication:

Major comments:

1.     Figure 1: Only lysates of insect stage T.  brucei or Leishmania parasites were probed with 6B6 by Western blotting and ELISA. This should be made clear in the text rather than hidden in the methods section. Also, given that the point here is to say that Tc323 is completely absent in T.  brucei and Leishmania (and therefore a good serodiagnostic marker in areas where T. cruzi and Leishmania infections coexist), then it would be good to show that 6B6 does not recognize proteins in mammalian stages of these other parasites. Also, only Leishmania mexicana lysate is assayed. Does 6B6 recognize proteins in other strains (also see below for a similar comment re. the phylogenetic analyses).

2.     Figure 2: What were some of the other proteins identified via immunoprecipitation and mass spectrometry? How many peptides were identified in comparison to Tc323?

Was only Leishmania major bioinformatically interrogated for Tc323 homologues as suggested in the figure? If so it would be nice to see a broader search of Leishmania species, especially given the Western blot and ELISA data in Figure 1 was obtained with L. mexicana lysate.

3.     Figure 3D: The presence of a signal peptide suggests that Tc323 localizes to the secretory pathway. What intracellular membranes is Tc323 associated with e.g., the authors could carry out immunofluorescence using the anti-BiP antibody for ER membranes, or also for Golgi markers such as TcAP1-gamma, TcHIP, or TcCLC for which antibodies exist in the literature. If possible, I would also like to see staining using an isotype control chimeric antibody against a target with alternative localization and different scFv but the same constant chains.

Is it surprising that patient serum and B cell antibodies are generated against an intracellular protein? Does this potentially suggest that Tc323 is completely secreted from the parasite, and antibodies are raised against the extracellular protein during infection?  

4.     Section 2.4: It is confusing to me that the simulation of antibody–antigen interactions is described in a section entitled ‘Tc323 is recognized by plasma from patients with chronic Chagas disease’. This could be better served with an alternative title, and also more description of what the docking tells us re. the immunogenicity of Tc323.

Also, in the description of the antibody–antigen interaction simulation results, the authors should explicitly make the conclusion that 6B6 appears to recognize a conformational epitope (currently hidden in the discussion).

Comments on the Quality of English Language

Care should be taken to assess language throughout as I detected several cases where it could be improved:

1.     Page 4 line 147, and page 5 line 163/164: ‘homologues’ instead of ‘homologous’.

2.     Page 8 lines 224-227: ‘Negative Z-scores are postulated as being better than the fitness of the HADDOCK cluster. Taking into consideration, the selected cluster presented high negative values of HADDOCK and Z-scores (Inner table from Fig 4A)’.

The language here could be improved but is also redundant with previous language, so I suggest deleting these sentences.

3.     Page 11 line 326: ‘Interestingly, Tc323-positive samples derived from Argentina, Bolivia, and Paraguay where prevail the genotypes TcV/TcVI, TcV and TcIII/TcV, respectively’. The language here could be improved e.g., ‘Interestingly, we used Tc323-positive samples derived from Argentina, Bolivia, and Paraguay, where the genotypes TcV/TcVI, TcV and TcIII/TcV, respectively are most frequent’.

Author Response

Reviewer #2

We express our gratitude to the editors and reviewers for their thorough examination of the manuscript and their constructive feedback. We have carefully considered their comments to improve and clarify the manuscript. Below, you will find a comprehensive point-by-point response to all the comments. All page numbers refer to the revised manuscript file with tracked changes.

  1. Figure 1: Only lysates of insect stage T. brucei or Leishmania parasites were probed with 6B6 by Western blotting and ELISA. This should be made clear in the text rather than hidden in the methods section. Also, given that the point here is to say that Tc323 is completely absent in T. brucei and Leishmania (and therefore a good serodiagnostic marker in areas where T. cruzi and Leishmania infections coexist), then it would be good to show that 6B6 does not recognize proteins in mammalian stages of these other parasites. Also, only Leishmania mexicana lysate is assayed. Does 6B6 recognize proteins in other strains (also see below for a similar comment re. the phylogenetic analyses).

Response: Thank you for pointing this out. The statement regarding which developmental stage was assayed for each species is now clearly indicated in the Results section of the revised version (Lines 99-10 and legend of Figure 1).

Also, in agreement with the Reviewer, we have included a western blot showing the lack of reactivity of scFv 6B6 against lysates from bloodstream forms of Trypanosoma brucei and promastigotes from various Leishmania species (L. infantum, L. amazoniensis and L. braziliensis). This result is now included in Figure S1. As expected, scFv 6B6 did not recognize any parasite stage or strain other than T. cruzi. This is because T. brucei, as well as any species from the Leishmania clade, lack Tc323 homologous sequences. Please see also the response question # 2 for an expanded explanation regarding the phylogenetic analysis.

  1. Figure 2: What were some of the other proteins identified via immunoprecipitation and mass spectrometry? How many peptides were identified in comparison to Tc323?

 Was only Leishmania major bioinformatically interrogated for Tc323 homologues as suggested in the figure? If so it would be nice to see a broader search of Leishmania species, especially given the Western blot and ELISA data in Figure 1 was obtained with L. mexicana lysate.

Response: In reference to the former question, we focused the mass spectrometry analysis on the identification of the protein band recognized by the scFv 6B6, which was always located at ~315-320 kDa in gels/western blots. Due to this, only the protein band located at this size (which was also stronger than others) was excised from the gel after IP and analyzed by mass spectrometry at the University of Dundee´s Facility. As a result, a total of 28 peptides (of them, 14 unique) with high Protein Scores matched with Tc323. Other 2 peptides matched with a flagellar attachment zone-related protein of ~180 kDa (Tc00.1047053509631.140) with a lower Protein Score. As the molecular weight of this protein did not match the size of the band recognized by scFv 6B6 and the immunostaining did not show a flagellum attachment zone-related location, we concluded these peptides were contaminants. 

In reference to the latter question, Tc323 homologues were searched in the 31 genomes of Leishmania strains sequenced to date and available on the TriTrypDB site, which included L. mexicana, L. infantum, L. major, etc. As a result, no Tc323 homologous sequences were identified in any Leishmania species. In the revised version, we added information in Lines 151-154 to improve the sentence and to better communicate this important finding.

  1. Figure 3D: The presence of a signal peptide suggests that Tc323 localizes to the secretory pathway. What intracellular membranes is Tc323 associated with e.g., the authors could carry out immunofluorescence using the anti-BiP antibody for ER membranes, or also for Golgi markers such as TcAP1-gamma, TcHIP, or TcCLC for which antibodies exist in the literature. If possible, I would also like to see staining using an isotype control chimeric antibody against a target with alternative localization and different scFv but the same constant chains.

Is it surprising that patient serum and B cell antibodies are generated against an intracellular protein? Does this potentially suggest that Tc323 is completely secreted from the parasite, and antibodies are raised against the extracellular protein during infection?

Response: Following the Reviewer's suggestion, we conducted immunofluorescence microscopy using the anti-6B6 in combination with anti-BIP or anti-GRASP antibodies as ER or Golgi markers, respectively (doi: 10.1074/jbc.M112.354696). These immunostaining studies showed that Tc323 does not exhibit a similar localization pattern to ER or Golgi membranes, suggesting that this protein may not be associated with these organelles (Figure S5). This result has been incorporated into the revised version of the manuscript. Additionally, concerning the staining of parasites using a different antibody with alternative localization but carrying the same constant chains, we have included in Figure S5 the staining pattern exhibited by the anti-A2R1 antibody (which recognizes tubulin from T. cruzi, DOI: 10.1016/j.ebiom.2020.103206). The A2R1 staining clearly differs from the Tc323 localization in the parasite body.  

In reference to the latter question, we consider it unsurprising that human plasma detects Tc323, as naturally occurring human antibodies to intracellular proteins are extensively described, especially in autoimmune diseases. In fact, different biological events expose many intracellular proteins to the cell surface or extracellular environment, such as necrosis or delivery. In the case of T. cruzi infection, several intracellular proteins have been tested and are currently used as serological markers for its diagnosis (DOI: 10.1590/S0074-02761999000700051). They include, to name but a few, CRA/ Ag30/JL /TCR27 (cytoplasmic proteins); TcE/JL5/TcP0 (ribosomal proteins); cy-hsp70/mt-hsp70/grp-hsp78 (heat shock proteins) (doi: 10.1016/bs.apar.2016.10.001; doi: 10.1590/0074-02760200444).

Having said that, the issue raised by the Reviewer is accurate and entirely feasible. As mentioned in the Discussion section, peptides from Tc323 were identified in the parasite´s secretome by other groups (doi: 10.1371/journal.pone.0185504). This might be due to Tc323 being secreted, thus enabling it to elicit a host humoral response. We thank the Reviewer for pointing this out. This is now emphasized in the Discussion of the revised version (Lines 318-319).

  1. Section 2.4: It is confusing to me that the simulation of antibody–antigen interactions is described in a section entitled ‘Tc323 is recognized by plasma from patients with chronic Chagas disease’. This could be better served with an alternative title, and also more description of what the docking tells us re. the immunogenicity of Tc323.

Also, in the description of the antibody–antigen interaction simulation results, the authors should explicitly make the conclusion that 6B6 appears to recognize a conformational epitope (currently hidden in the discussion).

Response: The title of this part has been modified in accordance with the Reviewer's suggestion and the description of docking analysis has been extended in Results and Discussion Sections.

Care should be taken to assess language throughout as I detected several cases where it could be improved:

  1. Page 4 line 147, and page 5 line 163/164: ‘homologues’ instead of ‘homologous’.

Response: Done

  1. Page 8 lines 224-227: ‘Negative Z-scores are postulated as being better than the fitness of the HADDOCK cluster. Taking into consideration, the selected cluster presented high negative values of HADDOCK and Z-scores (Inner table from Fig 4A)’. The language here could be improved but is also redundant with previous language, so I suggest deleting these sentences.

Response: We thank the reviewer for this comment. The sentences were deleted.

  1. Page 11 line 326: ‘Interestingly, Tc323-positive samples derived from Argentina, Bolivia, and Paraguay where prevail the genotypes TcV/TcVI, TcV and TcIII/TcV, respectively’. The language here could be improved e.g., ‘Interestingly, we used Tc323-positive samples derived from Argentina, Bolivia, and Paraguay, where the genotypes TcV/TcVI, TcV and TcIII/TcV, respectively are most frequent’.

Response: We appreciate your suggestion. The sentence has been changed accordingly.

Round 2

Reviewer 2 Report

Comments and Suggestions for Authors

The authors have answered all my points by addition of new data and editing of the manuscript.